# Best Practice Recommendations for the Implementation of a Digital Pathology Workflow in the Anatomic Pathology Laboratory by the European Society of Digital and Integrative Pathology (ESDIP)

**DOI:** 10.3390/diagnostics11112167

**Published:** 2021-11-22

**Authors:** Filippo Fraggetta, Vincenzo L’Imperio, David Ameisen, Rita Carvalho, Sabine Leh, Tim-Rasmus Kiehl, Mircea Serbanescu, Daniel Racoceanu, Vincenzo Della Mea, Antonio Polonia, Norman Zerbe, Catarina Eloy

**Affiliations:** 1European Society of Digital and Integrative Pathology (ESDIP), Rua da Constituição n°668, 1° Esq/Traseiras, 4200-194 Porto, Portugal; filippofra@hotmail.com (F.F.); vincenzo.limperio@gmail.com (V.L.); david.ameisen@gmail.com (D.A.); rita.carvalho@charite.de (R.C.); sabine.leh@helse-bergen.no (S.L.); rasmus.kiehl@charite.de (T.-R.K.); mircea_serbanescu@yahoo.com (M.S.); daniel.racoceanu@sorbonne-universite.fr (D.R.); vincenzo.dellamea@uniud.it (V.D.M.); apolonia@ipatimup.pt (A.P.); norman.zerbe@charite.de (N.Z.); 2Pathology Unit, “Gravina” Hospital, Caltagirone, ASP Catania, Via Portosalvo 1, 95041 Caltagirone, Italy; 3Department of Medicine and Surgery, Pathology, ASST Monza, San Gerardo Hospital, University of Milano-Bicocca, 20900 Monza, Italy; 4Imginit SAS, 152 Boulevard du Montparnasse, 75014 Paris, France; 5Charité–Universitätsmedizin Berlin, Corporate Member of Freie Universität Berlin and Humboldt-Universität zu Berlin, Institute of Pathology, Charitéplatz 1, 10117 Berlin, Germany; 6Department of Pathology, Haukeland University Hospital, Jonas Lies Vei 65, 5021 Bergen, Norway; 7Department of Clinical Medicine, University of Bergen, Jonas Lies Vei 87, 5021 Bergen, Norway; 8Department of Medical Informatics and Biostatistics, University of Medicine and Pharmacy of Craiova, 200349 Craiova, Romania; 9Sorbonne Université, Institut du Cerveau—Paris Brain Institute—ICM, Inserm, CNRS, APHP, Inria Team “Aramis”, Hôpital de la Pitié Salpêtrière, 75013 Paris, France; 10Department of Mathematics, Computer Science and Physics, University of Udine, 33100 Udine, Italy; 11Ipatimup Diagnostics, Institute of Molecular Pathology and Immunology of Porto University (Ipatimup), 4200-804 Porto, Portugal; 12Medical Faculty, University of Porto, 4200-319 Porto, Portugal

**Keywords:** digital pathology, anatomic pathology workflow, whole slide imaging, laboratory information system

## Abstract

The interest in implementing digital pathology (DP) workflows to obtain whole slide image (WSI) files for diagnostic purposes has increased in the last few years. The increasing performance of technical components and the Food and Drug Administration (FDA) approval of systems for primary diagnosis led to increased interest in applying DP workflows. However, despite this revolutionary transition, real world data suggest that a fully digital approach to the histological workflow has been implemented in only a minority of pathology laboratories. The objective of this study is to facilitate the implementation of DP workflows in pathology laboratories, helping those involved in this process of transformation to identify: (a) the scope and the boundaries of the DP transformation; (b) how to introduce automation to reduce errors; (c) how to introduce appropriate quality control to guarantee the safety of the process and (d) the hardware and software needed to implement DP systems inside the pathology laboratory. The European Society of Digital and Integrative Pathology (ESDIP) provided consensus-based recommendations developed through discussion among members of the Scientific Committee. The recommendations are thus based on the expertise of the panel members and on the agreement obtained after virtual meetings. Prior to publication, the recommendations were reviewed by members of the ESDIP Board. The recommendations comprehensively cover every step of the implementation of the digital workflow in the anatomic pathology department, emphasizing the importance of interoperability, automation and tracking of the entire process before the introduction of a scanning facility. Compared to the available national and international guidelines, the present document represents a practical, handy reference for the correct implementation of the digital workflow in Europe.

## 1. Introduction

The interest in implementing digital pathology (DP) workflows to obtain whole slide image (WSI) files for diagnostic purposes has increased in the last few years. This is due to the opportunities offered by WSI, e.g., telepathology and image analysis, including computational pathology tools based on artificial intelligence [AI] methods. The increasing performance of technical components and the Food and Drug Administration (FDA) approval of systems for primary diagnosis [1] led to increased interest in applying DP workflows. Moreover, in the last few years, several studies evaluating performance demonstrated the non-inferiority of WSI compared to conventional light microscopy [2,3,4] for primary histological diagnosis. This may help to alleviate concerns about the possible risk of DP-related diagnostic errors [5]. Indeed, the restrictions suffered during the COVID-19 pandemic, the reduction in the number of pathologists and the increase in workload, with rising number and complexity of cases, also raised the interest in DP. Several definitions for DP have been proposed so far [6,7], a common opinion being that DP encompasses the photographic documentation of the macroscopy of the specimens (“gross pathology”), the digitization of glass slides (virtual microscopy) and telepathology. By some definitions, DP involves merely the digitization of glass slides. In this study, “DP” is significantly distanced from the reductive paradigm of only glass slide digitization, moving towards a more integrative approach that comprises interventions in all stations of work in the pathology laboratory, introducing and supporting innovation. DP implicitly consists of all the associated technologies to allow improvements and innovations in workflow, including, for instance, laboratory management systems (LIS), digital dictation, dashboards and workflow management, electronic specimen labelling and tracking, and synoptic reporting tools. The objective of this study is to facilitate the implementation of DP workflows in pathology laboratories, helping those involved in this process of transformation to: (a) identify the scope and the boundaries of the DP transformation; (b) introduce automation to reduce errors; (c) introduce appropriate quality control to guarantee the safety of the process and (d) implement the hardware and software needed to implement DP systems inside the pathology laboratory. Since several recommendations and guidelines have already been proposed, primarily focusing on the validation of WSI for clinical purposes or on the technical environment, this paper mainly covers DP implementation and all the prerequisites for a pathology laboratory to change from an analogue to a digital workflow [8]. Considering all that has been reported about DP workflow implementation and its associated benefits, it is anticipated that this new methodology has many advantages that should be attractive and convenient for all pathology laboratories worldwide, independently of their dimension, workload, number of pathologists or type of activity (academic/nonacademic, private/public) [6,7,9,10,11].

So far, there are several possibilities to transit and to manage “images” in a digital workflow: an LIS-based approach [12,13], a scanner vendor approach [7] or an intermediate software approach (e.g., Linköping University [14]). Independently of the type of strategy chosen to switch towards a digital visualization of images (LIS-centric, vendor based or third-party software), the new system should be able to integrate every possible instrument (e.g., one or more scanners from same or different vendors with the possibility to manage different images from a variety of sources), preferably associated with a tracking system because of automation and innovation. The cost-effectiveness of DP has already been documented in implementation models that discuss the scope of investment, the potential return on investment, and cost-savings of DP, as well as any proposed income deriving from the adoption of WSIs [15]. Moreover, the adequate adaptation of a routine clinical workflow can finally lead to an optimization of resources (e.g., space, time, personnel, and equipment). These are intended as recommendations and suggestions for the implementation of the full DP workflow in the routine clinical practice of anatomic pathology laboratories. The introduction of a DP workflow even allows the implementation of computational pathology tools, i.e., artificial intelligence (AI). The following sections explain, point-by-point, the steps needed for the progressive, secure, and efficient transition into a DP workflow. Regarding cytopathology, there are several barriers that still need to be overcome for routine cytopathology to go digital and support wider adoption and sustainability. Therefore, the present study mainly focuses on histopathology and its transition to the DP workflow Box 1.

Box 1Digital pathology workflow implementation—Step by Step.
**Summary**
**Digital pathology is pathology—**A holistic approach that comprehends interventions in all stations of work at the pathology laboratory, introducing innovation.
**Digital pathology is attractive and convenient for pathology laboratories worldwide.**

**Digital pathology represents a safer and more efficient way of working and should be considered the new standard in pathology.**

**Implementation of a digital pathology workflow is the milestone to fully benefit from the potential of WSI and a prerequisite for the application of AI in routine diagnostics.**


## 2. Involvement of the Team in the Digital Pathology Transformation of the Laboratory

The implementation of digital pathology requires a multidisciplinary approach from the very beginning. The leading team should involve in-house participants (pathologists, laboratory technicians, administrative staff) and the hospital’s IT and technical services [6]. IT services might be organized in different ways depending on the size of the department and depending on local or national policies. For example, the IT services may be provided by individuals, by a separate department or by a subcontractor. The most important thing is that these groups work together and that they form a team. Subsequently, close collaboration with companies providing the digital pathology system and the laboratory information system will become necessary. Especially in larger departments, digital transformation will usually be organized as a project that includes a project manager, a steering group and different working groups. There are several ways of introducing the topic and designing the appropriate options for the laboratory at hand, and it might be useful to visit pathology departments with digital workflows to learn from their successes and failures. There are a couple of papers that share experiences and provide valuable information [6,7]. Describing user scenarios is another method to understand the needs of one’s own laboratory and communicate these to the IT and technical departments and possible suppliers. In addition, before starting a tender, it is helpful to gather information about suppliers and products. To obtain a successful implementation of the “DP” and to avoid deficiencies, the multidisciplinary team that is going to lead the “digital revolution” in each department should follow some crucial steps, as previously reported. In particular, for the correct and rapid implementation of DP in every department, it is advisable to create awareness, participation, appropriate work conditions, communication among the team members, and monitor the outcomes of this revolution. This approach could help in facing the heterogeneous patterns of reactions that different actors of the team could express, including the “enthusiasts”, the “sceptics”, and the “undecideds”. All the possible measures to increase the trust and involvement of pathologists should be applied to all staff members. To establish a successful DP workflow, a thorough stakeholder analysis should be carried out, and a communication strategy should be established based on this analysis. The team must ensure that all internal stakeholders (pathologists, laboratory personnel and administrative staff) are continuously informed from the beginning. In this setting, sharing the vision of DP with laboratory and administrative personnel, encouraging them to provide feedback, expressing potential concerns and suggestions (e.g., using frequent meetings on-site) and providing appropriate discussion during all phases of the deployment will facilitate a safe and effective implementation. The team must be aware that DP should be perceived as an integral part of the laboratory workflow rather than an “add-on” [6]. The contingent situation due to the COVID-19 pandemic can be further leveraged to boost the implementation of DP in the laboratories, stressing the need to maintain pathology services by making it possible for pathologists to work from home [16]. Implementing DP as the standard laboratory practice requires learning new technical skills to capture all the advantages of this technology. Just as significant as the internal stakeholders is the involvement of IT services. IT will be crucial in many aspects of the project (LIS adaptations, integrations, storage, testing etc.). The involvement should start in the early phases. For example, consider a laboratory office tour to establish communication with the other components of the project in clear language, understand what is expected and what is potentially achievable from your deployment, and what each professional group will be expected to contribute in terms of time and staff. Explain your ideas for future digital workflows and see what potential dependencies and solutions your IT colleagues can generate.

## 3. Optimization of Resources in the DP Workflow

In a fully digital laboratory, the processes and records are electronic file-based, the environment is paperless, with glass slides being substituted at the end of the workflow by WSIs. The optimization of resources, namely time, space, people and instruments, creates conditions for increased efficiency and, consequently, decreased costs. The LEAN approach represents a valuable strategy to optimize the workflow, leading to a more logical distribution of the spaces to minimize staff and sample traffic inside the laboratory. It also allows for a more harmonic and well-planned articulation between human resources and available instruments, which results in time and cost-effectiveness. Although it is not a strict prerequisite for adopting DP, it could further allow for better allocation of resources [17]. This can start from a more rational disposition of the spaces/offices inside the pathology laboratory. An inefficient arrangement of the physical spaces, typical of the old, “analogue” workflow, can partly impair the smooth crosstalk among the different components of the process. Previous experiences in implementation models stress the need to analyze the pre-existing workflow before implementing DP [6,7]. A careful analysis of the pre-existing analogic workflow before the transition should consider the flow of the samples (workstation location) in the laboratory and time intervals (hands-on and waiting times) for each workstation, verifying the information technology support and establishment of adequate quality control checkpoints. The lack of structural organization of some pathology laboratories, including the physical placement of the different workstations, may contribute negatively to the desirable, efficient crosstalk between workstations. The reorganization of such a laboratory structure with the intent to decrease unnecessary movements of the staff, and time loss, can be useful for every laboratory, independently of DP implementation. For instance, the scanning workstation should be located near the staining and mounting instruments, accelerating the production line but far from the microtome area to avoid the interference of paraffin with the scanning mechanisms. After this retrospective analysis and reorganization of the structure, the optimal choices for the automation of each workstation must be made, namely by the introduction of a reliable tracking system, and different instruments would preferably work in a coordinated fashion, connected (mono-or bi-directionally) to the LIS (or LIMS).

## 4. The Role and Potentialities of Laboratory Information (Management) System (LIS/LIMS) and Informatics Resources

Independently of the system employed to manage the WSI (LIS, scanner or third party), pathology laboratories mainly depend on laboratory information systems (LISs) to support their operations and, ultimately, carry out their patient care mission. For these reasons, one of the crucial points is to ensure the full integration of the systems involved in the digital transition. Although many LISs have evolved with sophisticated and more user-friendly software over the past few decades, supporting a broader range of functions, many others have not evolved, thus preventing possible integration with other technologies deployed in the laboratories. Modern LISs play different roles in all phases of patient testing, including specimen and test order entry, specimen processing and tracking. They track and organize the laboratory’s workflow, mainly through event logs and histology protocols. The maintenance of such logs can follow the default configurations of the system or can be customized by each laboratory to display the most useful information. A typical example of the system’s default configuration for a log (e.g., routine histology) includes accession number, timestamp, patient and specimen data, histology protocol(s) ordered, other stains ordered and comments about the specimen or the request. LISs now incorporate multiple features that, until recently, were either unavailable or required a significant customization effort to be obtained. The Association for Pathology Informatics produced a comprehensive list of basic and advanced LIS features that may be used to evaluate LIS capabilities [18]. Moreover, the next generation LIS should be able to link digital images to the respective cases appropriately. With the rising use of whole slide imaging (WSI) for clinical purposes, a consensual increase in capabilities to connect and integrate WSI systems and LIS is to be expected (e.g., open WSI from the LIS, log the viewed areas/magnification on all WSI or even apply image analysis and store result data). Further advances in the development of LISs are expected in the future, starting from the integration of more sophisticated tools to support data mining and the analysis of pathology and clinical data sets. The LIS may evolve into a multimodality “pathologist cockpit” that not only provides LIS functions but also displays pathology imaging and other medical imaging, supplies analytical tools, provides access to clinical data (e.g., Electronic Health Record [EHR]) [19] as well as other data sources [20]. A more recent guideline paper [21] underlined the importance of digital pathology interoperability, with a LIS being able to connect all the instruments present in the laboratory to support critical DP use cases. Moreover, increasing requests for molecular and genetic tests on pathology specimens (e.g., next-generation sequencing) impose further innovation in LISs to integrate and optimize these data with the traditional pathology report for optimal patient management [22] in an integrative model. Finally, the transition will allow information integration from grossing, enable collaborative work and incorporate quality control results.

## 5. Automation of Workflow and Tracking System

Automation and using a robust tracking system can significantly reduce errors related to handwriting transcription and misspelling that can cause samples to be dissociated from a particular patient (“mismatching”). Automation is a “strong recommendation” emanating from these recommendations, as it can benefit both pathology laboratories using DP and those using glass slides for diagnosis. Besides the introduction of a suitable LIS/LIMS that can help monitor the instruments’ performance connected to each sample, further automation can be introduced in the workstations. This includes the reagents used and tracking all the staff that were at any point involved in sample processing by differential log-ins or scanning of individual ID codes at all workstations. The possible automation of workstations obviously depends on budget, existing instruments, and the experience of the technical staff. Devices such as a robotic stainer and a cover-slipper will bring consistent slide quality, avoiding frequent re-staining and ongoing readjustments to scanning protocols. The same is true for the automation of embedding and cutting processes, for which available systems on the market appear promising. However, these are not yet widely used in practice [23]. The goals of a tracking system are to keep the sample automatically, correctly, and permanently labelled during the time that it circulates in the laboratory. The identification of the sample, using labels on the containers, printed in the cassettes/paraffin blocks, printed on the glass slides and then present in the WSI files, is a best practice rule that is recommended to be adopted for the use of the WSI. In this setting, the perfect compatibility (interoperability) of the instrumentation used to label and to process the samples within the AP laboratory, and with the other laboratories in the same institution, is crucial to avoid possible issues (e.g., blurring or shading of the labels during subsequent processing of specimens/slides). The sample identifiers, of which there are usually several (see Section 5), should be managed automatically and electronically connected to the patient’s LIS entry. The integration between the tracking system and LIS with an electronic interface between the LIS and the printers is essential to maintaining continuity of identification. The link established between the asset (tissue container/cassette/block) and the LIS will help reduce errors and can be achieved by printing different data types on the assets, such as barcodes or 2D (QR) codes. These can be linked to different types of data in the LIS. Eventually, other systems with code reader compatibility will be able to read them [24]. The introduction of radiofrequency identification (RFID) technologies is a promising method to track the assets, although cost and system integration barriers still limit their implementation [25,26]. In the case of pathology laboratories, introducing at least one barcode reader per workstation is recommended. Tracking an individual sample with the combined use of printers and code readers accelerates the work at the microtome stations, helping histotechnologists track each block and slide, ensuring the adequate identification and concordance between the individual block and slide labels [27]. The LISs typically offer the laboratories some capability to customize the format and content of their slide labels. As will be further explained in the subsequent sections of the document, the employment of unequivocal 2D barcodes can have a multitude of applications in the proposed digital workflow, significantly reducing the operations time and error rates. The impact of such implementation can be noted starting from the accessioning phases, where the sample is assigned its unequivocal code that will be used later during the processing and reporting steps. This can further help in the creation of tissue cassettes, in the production of tissue glass slides, in the automatic request of additional histochemical and immunohistochemical (IHC) stains, as well as in the double check that should be carried out at every checkpoint to ensure correspondence among received material, grossed specimen, embedded sample and cut sections. This is facilitated by the additional use of barcode readers and by the implementation of newly introduced instruments to capture the cut surface directly from the paraffin block [28], which is at this point essential to guarantee a sustainable and reliable quality control process (see Section 5).

As will be further explained in the following sections of this paper, the use of unequivocal 2D barcodes can have many applications in the proposed digital workflow, significantly reducing the operations time and error rates. The impact of such implementation can be noted starting from the accessioning phases, where the sample is assigned its unequivocal code that will be used later during the processing and reporting steps. This can further help in the creation of tissue cassettes, the production of glass slides, automatic requests for additional histologic and IHC stains, as well as in the double-check that should be carried out at every checkpoint to ensure the correspondence among arrived material, grossed specimen, embedded sample and cut sections. This is facilitated by the additional use of barcode readers and by implementing instruments capable of capturing the cut surface directly from the paraffin block (see Section 5).

## 6. Quality Control Program and Definition of Checkpoints

Quality control of products from a pathology laboratory is essential to guarantee that a patient receives a correct diagnosis. In Europe, the certification and accreditation of laboratories are not equally and uniformly performed across the territory. Instead, many laboratories design their own quality control program, more or less simplified, often involving only segments of sample processing adequate to their intent. Although adopting a quality management system is not strictly required in all countries as a prerequisite for implementing DP workflow, laboratories with a robust system of quality management may find the DP workflow easier to implement as they are already aware of the critical control checkpoints through the analogue workflow. To support those laboratories that are not yet familiar with quality control programs, a detailed description of some suggested checkpoints suitable for adaptation to each laboratory are provided. The checkpoints described here derive from the need to control the performance of a new instrument in the pipeline—the scanner. They also originate from introducing new standard operative procedures (SOPs), tools/instruments and quality control of the processes (Figure 1).

Simultaneously, per each workstation, some technical modifications are discussed to facilitate the scanning process and increase the quality of the WSI. We highlight that time loss within the laboratory is frequently motivated by a mismatch of samples and poor sample quality (either due to a pre-analytical or analytical factor). Investing in a workflow with a good quality of samples that are easy to track decreases the time lost, considering that this loss is very difficult to estimate because it is not generally recorded.

### 6.1. Accessioning Checkpoints

During the accessioning phase, samples that arrive at the pathology laboratory are registered in the LIS and given a case number. Analyzing the classical analog accessioning procedures allows for a critical evaluation of the potential issues that can and do happen. Mistakes can occur in the compilation of paper requests from the submitting department or outside hospital (for internal and external cases, respectively). The staff responsible for the accessioning phase can miss discrepancies between the sample/slide and the request or even mismatch this pair. The manual insertion of the specimen/patient data into the LIS can impair the link with the patient profile present in the hospital information system (HIS), generally due to an inappropriate transcription of a patient’s identification data, eventually causing a duplication of patients’ profile, and consuming time. In a laboratory with a DP workflow, laboratory personnel have the possibility of completing these accessioning tasks automatically to minimize the risk of errors. The different identification codes (IDs) used in the various subsequent steps play a crucial role here. Similar to a pyramid or hierarchy, different types of IDs are attributed to the patient, and everything associated with this accessioning event, as follows:Patient IDC IDSpecimen container ID (entry lab)Sample IDsBlock IDsSlide IDs

For DP, a mechanism is needed to get the following types of required case information to the administrative and the pathologist: patient ID and demographic information, description of the specimen, clinical history and questions or requests for the pathologist. In the digital pathology laboratory, the accessioning is modified and may include some or all of the following (Table 1): Sample/slides arrive in the pathology laboratory with a label containing a code (entry lab, preferentially 2D type) associated with patient and case data.By scanning the code on the label of the case, the administrative is able to open the digital request on pathology LIS automatically, allowing the automatic synchronization of the information from the hospital system or creating the specific page for cases/patients coming from outside.A case ID for the sample is generated.The case ID is then used in all sorts of assets generated for that case (cassettes, new slides, special stains, digital slides).The administrator can take pictures of both the container and the specimen, and these photos will be attached to the case file.All of the documents received together with the specimen are scanned and attached to the case file or directly transmitted to the LIS digitally (Optical Character Recognition, OCR).

### 6.2. Grossing Checkpoints

After accessioning, cases are ready to be macroscopically analyzed, described and grossed by the pathologist or trained technical staff. As this happens in accessioning, the grossing workstation may be a source of human errors. These errors may include some of the following: wrong assignment of the macroscopic description and grossing of one patient in the paperwork of another patient, loss of manual transcription of specimen descriptions, deterioration of the numbers on the cassettes and incongruences among the sample received, grossed and subsequently processed in the absence of step-by-step picture documentation.

As in the other workstations, through automation the DP workflow can help reduce to a minimum the human interference needed in the grossing phase. The previously described process would be as follows (Table 2): The grossing operator (e.g., pathologist/resident/pathologist’s assistant) can access the case/patient file by directly scanning the code on the sample container.Pictures are taken of the sample before it is described and grossed, as well as during grossing, and finally of all tissue cassettes with slices; those images are directly linked to the case using the software integration paths between the LIS and the image capture instrument.The grossing operator performs a macroscopic description of the sample through automated speech recognition systems that report the text in the appropriate section of the case/patient file using the software integrations paths between the LIS and the dictation system instrument.The operator can produce cassettes by using a specific printer (preferably laser printer) to assign an identification code corresponding to the particular case, as established during the accessioning and using the software integration paths between the LIS and the printer. The cassettes and marker media should be appropriately tested to demonstrate the indelibility or impossibility of washing away or removing the identification code. The suggested code is 2D (e.g., QR code), which can include a greater character count (higher data density), require a smaller footprint, and have fewer scan and printer failures than 1D codes.An image of the cassette with the grossed specimen should be obtained at the bench, allowing retrieval of this at the following steps.

### 6.3. Grossing-to-Processing and Processing Checkpoints

After the grossing phase, cassettes containing the specimens are ready to be processed. At this stage, further checkpoints may be needed to verify that all the cassettes generated at the grossing workstation are present in the rack to be processed. This double-check is still routinely and primarily done manually in most pathology laboratories. In a DP workflow, this task may be carried out by scanning the codes printed in the cassettes of the rack before they are processed and checking if all the produced cassettes are submitted to the subsequent phase, integrating the information in the LIS. During the processing phase, both the instruments and programs used should be preferentially recorded through the employment of an appropriately integrated LIS, allowing for tracking the specimens/cases at this workstation. This system can be further deployed to track the usage of reagents for processing, helping in the safe disposal of these reagents. Moreover, the integration with the LIS can further help aggregate specimens and cassettes in different racks based on their processing time and scanner time/protocol requirements (e.g., fast vs. standard processing), or even separate specimens processed in different instruments.

### 6.4. Embedding Checkpoints

Once the processed specimen inside the cassettes have arrived at the embedding room, operators (technicians) should be able to access the pictures taken during the grossing phase by simply scanning the barcode (Table 2). This will allow them to compare them with the content of cassettes after processing, checking their correspondence to rule out the loss of biological material. Correct embedding may prevent the creation of poor-quality virtual slides. One of the possible issues during the scanning phase is represented by the presence of large fragments, which are more prone to be hydrated during the processing steps and thus more complicated to be captured by the scanner. To address this problem, the fragments should be reduced during the grossing phase, and the embedding checkpoint is essential to control this point. Similarly, tissue fragments well oriented, levelled, and close to each other in paraffin, may constitute good substrates for better-quality glass slides. If the sample to be analyzed is too large to be fitted in a regular glass slide, the recent introduction of dedicated scanners for “macro” glass slides provides the possibility of this solution directly from the grossing room [29].

### 6.5. Sectioning Checkpoints

The sectioning workstation is a time-consuming phase of the laboratory flow where errors are frequent. Here, the automation can facilitate the technician’s work bringing increased control, fewer errors, and resulting in less time spent. The sectioning workstation is complex and requires the rapid manipulation of specimens and instruments in a consecutive way. The introduction of a slide printer, a code reader, a desktop interface, and similar devices can be initially perceived as a further complication of this step. Checkpoints can be installed at this workstation depending on the laboratory’s needs and may prevent important errors (Table 3). Moreover, the employment of a slide printer (e.g., laser) at the sectioning station connected to the tracking system should be preferred on the “classic” handwritten or printed labels to reduce the risk of mismatches. The LIS should also be the source of all the information regarding the types of stains to be performed (i.e., IHC or “special” tinctorial stains) from a specific block. Moreover, the introduction of dedicated instruments to capture the cut surface of each paraffin block [28] could represent an additional checkpoint, helping to further reduce tissue inconsistencies among the blocks and the final glass/virtual slides. The sectioning process should follow the highest operative standards to minimize errors and poor quality in the subsequent scanning phase. Indeed, the irregular thickness of a tissue section, and the presence of holes or scratches and debris erroneously collected from the bath can impair the correct scan of the final glass slide product. The same is true for sections located at the edges of the slides, which may pass undetected by the scanner. Thus, the sections should be thin enough during the cutting phase and preferentially located in the middle of the physical glass slides to ensure the most appropriate scanning quality. Automatic microtomes may contribute to decreased tissue thickness variations.

### 6.6. Staining and Mounting Checkpoints

Once in the staining workstation, the slides produced in the digital workflow are identified through their code to define which staining protocol they should follow, as per internal LIS prerecorded indications, using automated staining platforms. As in the previous step, the staining process should follow the highest qualitative standards to reduce possible modifications that can interfere with the scanning phase (faint or darker staining, debris/precipitates). For this purpose, implementing an internal checkpoint with daily controls and/or external quality control can help assess the quality of stained slides [30]. Automating the staining may contribute to a stable result, allowing the design of a scanning protocol applicable to most of the slides, avoiding restaining and rescanning slides. The production of consistent staining with a clean background is relevant because it decreases the size of the produced digital slides. A final word is needed to address the mounting process and respective automation. To minimize the interference of the mounting medium in the scanning process, the laboratory must select the mounter, the coverslip type, and respective mounting medium to be used in all sorts of glass slides so that the scanner can be calibrated accordingly. Before the scanning phase, it is of paramount importance to check whether the slide is in an adequate state for scanning. After the staining/cover-slipping phase, it should be dry to prevent scanning problems (e.g., stitching, blurring, out of focus areas). Moreover, the scanning phase can be either affected by the different positions of the coverslips, leading to a misalignment of the slides in the rack. Differences in the type of coverslip can be responsible for a high rate of WSIs being out of focus. The use of automatic mounters obviates variations in the quality of the mounting and prevents errors if an adequate revision of the mounter is provided.

### 6.7. Correct Assigning of the WSI to the Case Checkpoints

Please refer to Section 7 of the present document.

### 6.8. Archiving Checkpoints

After the sectioning and scanning phases, blocks and slides can be appropriately archived to be retrieved whenever is necessary. This task has been historically performed manually by operators (technicians or laboratory assistants), leading to loss and misplacement of blocks/slides, with obvious medico-legal consequences. Moreover, the wide practice of consulting archival material by all the laboratory workers, including residents and students for didactic purposes, can further complicate the correct positioning of these specimens. Based on these observations, the full integration with the LIS and the presence of unique identifiers, both on the blocks and glass slides, allow an automated archiving of all the biological material, as well as its safe and unbiased retrieval if needed (e.g., request of external consultation). For archiving of the WSI, please refer to the data retention policy (Section 8 of the present document).

## 7. Scanner for Slide Digitization

This section contains some considerations and recommendations for selecting and managing the most appropriate digital slide scanner (Table 4). As in other medical specialties, which have been dramatically changed by the introduction of a wide variety of digital devices for the routine daily work [31], it is not the focus of the present recommendations to draw a meticulous review of the technical characteristics of a scanner, since several studies have already been published on this subject [32,33].

The most appropriate scanner should be selected based on the needs of the specific laboratory (e.g., primary diagnosis, consultation, education, and research). The following section focuses on the possible impact of such a choice on DP workflow implementation. According to the LEAN approach, as previously stated in chapter 2, the positioning of the scanners should follow the logic of an automated workflow and thus be placed as close as possible to the staining and cover-slipping stations, making their implementation in the entire process easier and smooth [6,7]. The transition to digital pathology also includes choosing the most appropriate types and numbers of scanners for the lab. Although it is highly dependent on the needs of each specific laboratory, one way to estimate the number of scanners is to review previous DP experiences [6,7]. In this context, each department should be aware of the expected application of the scanners, the total time required for the scanning process, and the time that can be dedicated to this part of the workflow. Some formulas to calculate the numbers of scanners needed in the lab have been proposed [6,7]. However, many variables must be considered when calculating the number of scanners required to digitize the entire slide volume within the same workday, thus not interfering with the TAT. These variables are limited by the technical specifications of the scanners and the informatics networks (including bandwidth and switches), type and location of storage, together with the existing workflow within the lab (i.e., availability of the personnel 24/7). One of the possible pitfalls in calculating the actual scanning time per slide/batch, and thus the number of scanners required per lab, could be represented by the reported scanning times by each vendor, generally calculated on a sample tissue of 1.5 cm × 1.5 cm in size and with a local storage solution. However, this is far away from the routine practice of an anatomic pathology laboratory that must accommodate very small pieces of tissue (e.g., biopsies) as well as large surgical samples, and that may even have the possibility of storing the WSI remotely or in the cloud. Moreover, since the implementation of scanners should not impact the existing workflow and eventually lead to its improvement, there is a need to evaluate a continuous loading capability to preserve the same or similar workloads over time compared to the conventional analogue counterpart. This should be coupled with the possibility of prioritizing a specific batch of slides.

However, a few comments are needed. Overall, scanning during working hours should be preferred for practical and logistical reasons. If there are problems with the scanning process, it might be better to avoid scanning after working hours. For example, it has been reported that the mean scanning time in a routine environment is about 6 min for scanning a slide at an equivalent of 40 × magnification [34]. Therefore, it takes about 4 h for one rack of 40 slides and up to 40 h to digitize all of the slides that fit inside the scanner (using an AT2, Leica Biosystems, Nussloch, Germany). However, it is well known that the scanning process may stop for several reasons, including sticky glass, connection problems or software and hardware problems. The result may be an incomplete digitization of slides, with consequent interruption of routine workflow in the subsequent morning. Based on the previous observations, and since the scanning process should be a continuous workflow in the lab, scanning during the day should be preferred to the overnight approach. This could enable lab personnel to react to the possible technical issues mentioned above. This could even lead to modifications in other parts of the pathology workflow to adapt routine specimen processing to the loading schedules required by the scanners. In this setting, the laboratory can choose to switch from bulk production of slides at the end of the day to a more continuous production of samples. Once the number of scanners needed and the required scanning time has been defined, the laboratory should verify whether the number of working operators employed in the department is sufficient to run the instruments for the specified length of time. Otherwise, the calculation of the necessary personnel is required, considering both:The scanning process.Virtual slide quality control.

The first part mainly consists of loading/unloading slides in the scanner and taking snapshots to ensure that the instrument captures all the material on the glass. On the other hand, the quality control phase is equally important and may be time-consuming, encompassing all the quality check procedures of the final WSI and related data. The points discussed above pertain to “regular” scanners for bright field microscopy. Other “special” scanners exist, e.g., those for dark field microscopy (immunofluorescence and fluorescence in-situ hybridization, FISH), as well as those for whole mount slides (macro slides). Because of their highly specific fields of application [35], they are not the object of these recommendations. 

## 8. Validation of WSI for Clinical Use

Several validation studies for WSI have been published, and most show broad concordance between the conventional microscope and the digital diagnosis [3,36]. Regarding staining and sample types applicable to WSI-based diagnosis, most basic tissue slides stained with hematoxylin and eosin (H&E), as well as most special stains and IHC stains, are expected to be usable. However, they require appropriate validation studies, followed by trial periods until the users have reached an adequate learning level. The recommended validation period for the clinical use of WSI should allow each pathologist to follow a training phase with parallel access to glass and digital slides for each case, with different wash-out intervals of time proposed.

This path for the implementation of WSI for primary diagnosis has been followed by different laboratories worldwide [8,37,38]. Some pathologists’ professional societies (e.g., College of American Pathologists) have proposed detailed guidelines for this validation process [32,39]. These have recently been updated, although they are mainly centered on validating WSIs in the diagnostic setting, not considering all the preanalytical phases of the digital workflow [40]. Here we discuss further critical points that have recently emerged as impactful in the implementation and validation of WSI. They include the most appropriate visualization devices, assessment of scan quality, tissue coverage of the block, glass slide and virtual slide and the proper assignment of the WSI to the case and/or the patient. 

### 8.1. The Visualization Chain: The Most Appropriate Monitor and Display. The Pathologist Workstation

The typical pathologist workstation is composed of one computer and two monitors. One monitor displays the LIS showing patient data and different dashboards with the possibility to access the patient’s documents or slides. The other monitor is dedicated to the visualization of the WSIs or other images. Several documents have already described all of the features needed to implement the visualization instruments in DP, namely, monitor quality, brightness and contrast, color depth, fidelity and profiles [32,41]. Many pathology departments already operate with workstations equipped with high-contrast (e.g., minimum contrast ratio of 1000:1), high–resolution (e.g., 16:10, 27′ diagonal matrix, 2560 × 1600), and bright displays (e.g., a maximum brightness of 300 cd/m^2^). Some FDA-approved built-in solutions for digital pathology employ medical-grade displays. However, the minimum requirements for DP monitors are still debated, and there is no consensus on how to assess their quality. External sources of variability further complicate this matter, such as the distance from the monitor and the illumination conditions of the room, which makes unbiased comparisons among the different devices available more difficult. This heterogeneity, and the large variety of supply in the digital market, has been recently reviewed [41], stressing the need for appropriate information of pathologists on this topic due to the complexity of the available technologies, which are changing at a fast pace. Alternatively, an easy-to-access and point-of-use quality assessment tool has been proposed, which tests color accuracy and may be a valuable indicator of the suitability of a particular screen for digital pathology diagnostics. Further validation is needed for its definitive employment in this setting [42]. Many comments could be made to identify the minimum computer technical requirement at the pathologist’s workstation. Dedicated random access memory (RAM) allows pathologists to visualize the WSI correctly. However, there is no standard in selecting CPU or RAM, as the management of WSIs may be affected by several parameters (i.e., network connection, switches etc.). A recent mid-range gaming computer will undoubtedly exceed the technical requirements to approach DP.

### 8.2. Scan Quality Assessment

After digitization, the produced WSI should be checked to ensure appropriate image quality to avoid any technical interference with the final diagnosis. Although most of the routine slides (about 90%), if properly processed, should not present scanning problems, some special slides (e.g., IHC or FISH) would benefit from dedicated scanning protocols and could be affected by more digitization issues. On the other hand, a minority of the “routine” slides (about 10%) could still be affected by scanning issues, stressing the need to adopt alternative protocols to obtain WSI from these challenging samples. This is based mainly on the assessment of focus quality, which can be partly assisted by the automated metric implemented in some available scan systems but should be performed on every slide to decide whether to rescan the sample. This can be done systematically by the lab personnel (e.g., technicians) for every scanning set before assigning the case to a pathologist. Alternatively, the pathologist can perform this check once after case review, requesting a rescan of a glass slide similar to the way that an additional recut or a special stain is ordered in the LIS. However, under real-life conditions, the entire manual check could be rather time-consuming and troublesome, especially in light of the need for subsequent deployment of image analysis algorithms. For this reason, automation of this phase is highly recommended. It can further speed up the digital transition process, ensuring an adequate quality of the scanned slide for potential subsequent AI analyses [43]. Suppose pathologists review a slide with blurry areas. In that case, it is up to their judgment to decide whether these artefacts will interfere with their safe diagnosis of the image, and order rescans as necessary. However, even in this case, it can be challenging for the human naked eye to unmask potential slight imperfections of the scanned slides that can impair the employment of AI algorithms. Even in these cases, the introduction of focus quality assessment [44] and quality control computational tools have been developed. In this setting, further validations are needed to implement such algorithms in routine practice [43,45].

### 8.3. Tissue Coverage

A critical assumption with using WSI in clinical settings is that scanned slides are completely accurate digital representations of glass slides. Therefore, it is of paramount importance that all tissue fragments present on glass slides be recognized and captured for review on the resulting digital slides.

Typically, two main different images are generated during the digitization process:The overview (rendering the macro/slide label files) that is a low-resolution snapshot of the entire glass slide.The “digital image” of the glass slide generated by a microscope camera (rendering baseline tiled image, thumbnail and multiple intermediate tiled images stacked in a pyramid) often acquired at the chosen magnification.

Many WSI devices include systems to detect tissue samples on a slide to limit scanning to relevant tissue-containing areas. For example, some scanners are programmed to omit the blank areas on the slide during the scanning process, where tissue is presumed to be absent, to speed up the process and generate files of smaller size. However, sometimes the tissue detection mechanism can fail to identify small or pale pieces of tissue automatically (e.g., label them as blank areas), or the user may not appropriately select the region containing the entire tissue analyzable. This may lead to potential errors that can cause serious discrepancies between the tissue present on the glass slide and the WSI. In this setting, overview images can help avoid such errors and represent a valuable tool for quality assessment purposes [46]. The macro image provides a low-magnification overview of all the tissue pieces and empty space on the glass slide. It serves mainly to guide the scanner’s tissue detection system, focus-point selection, and subsequent high-resolution digitization of tissue recognized and/or manually selected by an operator. However, the macro image is not necessarily displayed by default by all WSI vendors. Pathologists and laboratory personnel should be adequately trained on how to find and use the macro image as part of essential quality control. An alternative way to ensure that all the available tissue is present on the WSI is to compare the obtained digital slide with the original glass slide. However, this process could be time-consuming and represent a continuing additional workload for the lab personnel (e.g., histotechnologists), avoidable with the proposed double-check practice with the macro images made by each pathologist when the virtual cases are assigned [46].

Other possible sources of tissue coverage errors can result from inappropriate placement of tissue sections on the glass slide (e.g., below the label or on the external frame of the slide), which can fall outside the area recognized by the scanner. To address this issue, it is also essential to check the technical specifications of the scanner, specifically regarding the predefined detectable area that could be insufficient to cover the entirety of the physical glass slide containing tissue. These issues can be addressed by following the point-by-point indications reported above on the sectioning and cover-slipping checkpoints. Finally, very few cases with concordant WSI and macro images can still hide discrepancy issues with the entire amount of tissue sent for the analysis in the lab. This could be addressed by the manual, analog comparison of the tissue block with the macro images and the WSI, as suggested in the relative checkpoint in Section 5. However, the use of appropriate instruments to take a picture of the cut surface of the block can represent a valuable digital cost-effective alternative to reduce the error rate further. The pathologist can then readily check the three-way concordance among digitized cut surface of the block, macro images and WSI for every assigned case, reducing the error rate close to zero.

### 8.4. Assignment of Images to the Correct Case/Patient File

At the end of the scanning process, one of the most important steps is correctly assigning the digitized slides to the appropriate case/patient. As already mentioned, during the scanning phase a macro image is generated representing a snapshot of the entire glass slide that usually includes the slide label with identifiers (e.g., case accession number, barcode, text showing a patient name, and slide level or stain details). As per other steps mentioned in Section 5, a 2D barcode is crucial here as well to allow the scanning system, adequately integrated with the LIS, to link the scanned slides to specific specimens and patients. In some cases, according to institute policies, the dedicated personnel can perform a double-check after the automatic assignment of slides. It has been reported that errors in recognizing the printed barcode on the slide (or barcodes printed in the label) may occur, thus preventing the WSI to be matched with the corresponding case. This is due mainly to the poor quality of the printed barcode (or because it is missing a part of the barcode). This ultimately results in a case that is not ready to be reported, with a consequent delay in diagnosis. A checkpoint should be performed at this step to verify that all the stained slides have been digitized and assigned to the correct cases. Usually, scanner vendors create specific folders of “unassigned” (or barcode-less) slides for this purpose.

## 9. Open Topics (Not Fully Addressed in This Document)

### 9.1. Data Retention Policies and Image Storage Solutions

Data retention policies and image storage solutions for WSIs still represent open and debated topics, and no strict recommendations have been provided yet, although some suggestions and indications can be found in different documents and guidelines [32]. General recommendations, including those from the College of American Pathologists (CAP), currently advise retaining glass slides for at least ten years [47], with suggested periods of retaining digital pathology storage for a period ranging from a few years up to several years [48]. However, regulation on the retention of virtual slides, when used for primary diagnosis, is still lacking. Some documents suggest applying the same indications used for glass slides for WSIs, too [31]. Other recommendations are to keep the digital image for a period of two laboratory inspection cycles in case of any need to review it (e.g., for audits, quality control, medico-legal reasons) [32]. However strict and precise European guidelines are still needed to define the minimum period of time for WSI storage. Until then, pathology departments should determine an appropriate retention policy for the digital images [49].

Closely related to the retention policy is the appropriate storage of the images: where, how, and which slides must be stored is still an unresolved issue with multiple possibilities that can be adapted to each laboratory’s needs. Regarding the solutions available on the actual storage of the WSIs, organized and redundant storage solutions (e.g., Network Attached Storage, NAS, or Redundant Array of Independent Disks, RAID) are preferred to simple external/internal hard drives, considered negligent by some authors [31]. The possibility of multiple backup copies and disaster recovery procedures should also be kept in consideration [48]. Moreover, the issue related to the file extension of the WSIs that should be employed during the back-up is still unsolved. One of the essential requirements is represented by the capability of ensuring the changelessness, the guaranteed future and future-proofing of the data (e.g., the unity of patient or case data and the actual image content), as well as the easy accessibility of the WSIs, even after years. This can be obtained by the use of a DICOM-capable archive, although the eventual loss of quality related to the compression/conversion from one image extension to another is still a matter of debate. Finally, identifying the amount and type of storage needed is important, as it is one of the highest costs when implementing DP and needs to be adapted to the calculated yearly needs of each laboratory [15].

It has to be underlined that digital pathology storages need to be built to be interoperable and useful. This requires high-quality datasets, seamless communication across IT systems and standard data formats [50]. Interoperability is of paramount importance for achieving the full potential of digitization in healthcare and medicine to avoid the risk of having data difficult to exchange, process and interpret. Interoperability should be technical, syntactic, semantic and organizational [50]. In this line, it has been suggested that enterprise Vendor Neutral Archiving, composed of hardware and software, could be used to accumulate images directly from various image acquisition sources with the possibility to manage images and other end-user applications such as electronic health records, laboratory information systems, and other health-related information systems and databases [51].

### 9.2. Evaluation of the Results Obtained with the Digital Transition

The process of DP implementation has designs and consequences that are distinct in each pathology laboratory. Monitoring the effects of the digital transformation of the laboratory is good practice. It may include an analysis of the following parameters before and after the implementation: results of quality control, turn-around time, workload for technicians and pathologists, ergonomics of each workstation, and the general satisfaction of the staff. For validation and quality control of the implemented digital workflow, we direct the reader to the recommendations already in use [32,33,40].

### 9.3. Preparing for the Subsequent Steps after Implementing the Digital Workflow

Having a digital repository that contains clinical and histological information can lay the foundation for using numerous computational pathology tools. The application of image analysis algorithms can allow the identification of specific cell or tissue compartments (e.g., nuclei, mitosis, glands, stroma, among others) for quantification (e.g., cell or mitosis counting) as well as for classification purposes (e.g., grading) [52]. Some practical applications of these tools range from helping in the more rigorous scoring of some IHC-stained sections (e.g., programmed death-ligand 1 [PD-L1] and human epidermal growth factor receptor 2 (HER2) scoring) to the quantification of the proliferation index (Ki-67) in many neoplasms (e.g., breast, lymphomas and neuroendocrine tumours) [53,54]. Moreover, the recent introduction of more sophisticated elaboration algorithms allows further information starting from the digitized images and integrating the clinical, laboratory and radiological data to obtain diagnostic, classification and prognostic hints through the application of the so-called artificial intelligence (AI) [55,56]. Further simplifications in the work of pathologists are possible in the future, such as the automation of time-consuming repetitive tasks and the extraction of more data from the tissue to support precision medicine.

## 10. Closing Remarks

The present recommendations represent a European guidance for transitioning from a classic, “analogue” to a completely digital workflow in every anatomic pathology department. Ten basic principles (Table 5) resulted from the discussion among international experts after implementing the available updated national guidelines. Based on the present document, the anatomic pathology societies of every European country should be able to direct the departments towards DP transition. Updates in the future will provide dedicated indications on the adoption of computer-aided diagnosis and AI tools.

## Figures and Tables

**Figure 1 diagnostics-11-02167-f001:**
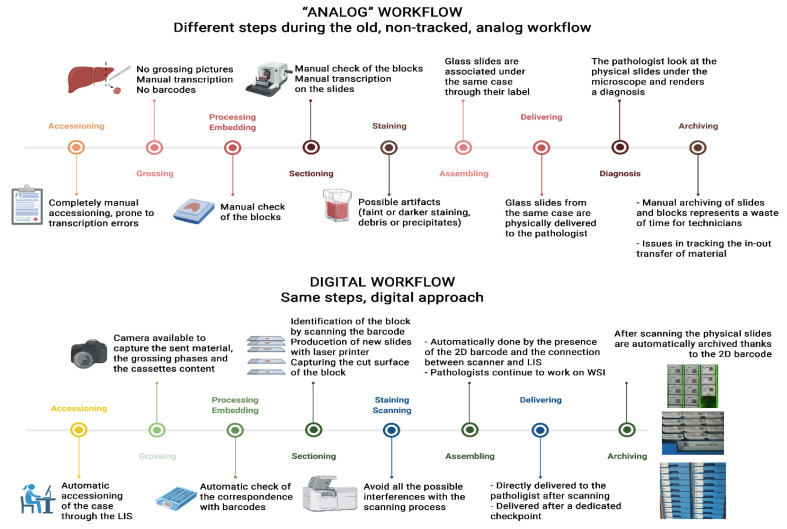
Differences among analog and the digital workflows. Credit: created with BioRender.

**Table 1 diagnostics-11-02167-t001:** Suggested checkpoints at the accessioning workstation.

Accessioning Checkpoints
Samples/slides are accessioned, using an order entry system, after the scanning of a code that identifies the patient/case, imports all the necessary information from the integrated HIS and opens the digital request: a procedure that introduces automation and consequent reduction of transcription errors.
2. A number and the respective identification code are generated for each sample, and the identification code is used for various assets generated for that case: procedure that allows the tracking of the sample while it is circulating in the laboratory.
3. Dedicated personnel take a picture of the container and of the specimen, and those photos are attached to the case file: procedure that documents the product entering the laboratory as well as the respective identification; this may represent an important medico-legal registry.
4. The documents that may be received with the specimen are scanned and attached to the case file or directly transmitted digitally (OCR): a procedure that facilitates access to relevant information that is prevented from being lost in a workstation.

**Table 2 diagnostics-11-02167-t002:** Suggested checkpoints at the grossing workstation.

Grossing Checkpoints
Scanning the identification code on the sample container allows for automatic access to the patient/case data, preventing transcription errors.
2. The photographic documentation of the sample as it is in the container, during grossing and within the cassettes (for comparison to what is arrived at the embedding station) guarantees the preservation of the case features and identification. The automatic introduction of the photographs into the patient/case file at the LIS prevents mismatches and time loss.
3. The macroscopic description of the sample is dictated and converted to text through voice recognition functions of the LIS or of an instrument connected to the LIS preventing transcription errors and time loss.
4. Cassettes are printed with the identification code of the sample to be tracked in subsequent workstations.
5. The material inserted in the cassette during grossing can be captured to obtain retrievable pictures at the following steps.

**Table 3 diagnostics-11-02167-t003:** Suggested checkpoint at the sectioning workstation.

Sectioning Checkpoints
The code printed on the paraffin block may be scanned to open the case file through the integrated LIS preventing transcription errors.
2. The technician can check how many and which kinds of slides are needed for each block directly on the LIS.
3. For each paraffin block, one or more printed glass slides are then generated through a dedicated printer, with all the slides having a unique identifier.
4. After sectioning, each paraffin block may be photographed to assess whether all the material emerged on the glass slide/WSI.
5. The sectioning phase should follow high operative standards, reducing the risk of artifacts that can impair the scanning phase.

**Table 4 diagnostics-11-02167-t004:** Recommendations for the scanning phase.

Scanning Checkpoints
At any given time, two scanners digitise twice the number of slides compared to a single scanner, and three scanners triple this (e.g., for a caseload of 300 slides per day, employing three scanners with 100-slide capacity could be better than using a single scanner with a 300-slide capacity).
2. It is advisable to scan during the daytime, with the lab personnel present to solve unexpected problems.
3. Scanning sessions during the night might be problematic in some already established workflows; thus, if there are problems with the scanning process, it might be better to avoid scanning after working hours.
4. A single-scanner approach is not recommended when contemplating a daily routine diagnostic workflow.
5. Consider the possibility of a continuous loading and eventual prioritisation of a batch of slides.

**Table 5 diagnostics-11-02167-t005:** Summary of the recommendations for the implementation of the digital workflow.

Principles	Type of Action
The transformation of a laboratory toward Digital Pathology requires a multidisciplinary approach (pathologists, technicians, IT).	Recommendation
2. Involve all the team in the transition process toward Digital Pathology (Educational phase).	Recommendation
3. Spare valuable resources (e.g., spaces, time and people) employing the LEAN approach to optimize the process.	Suggestion
4. Analyze the potentialities of the laboratory information (management) system (LIS/LIMS) and be aware of the information resources.	Recommendation
5. Start the automation of all the possible processes, implementing a tracking system and defining the appropriate checkpoints for every phase.	Recommendation
6. Design a quality control program mapping the necessary quality control steps.	Recommendation
7. Choose an appropriate scanner.	Suggestion
8. Validate WSI for clinical use.	Recommendation
9. Evaluate the impact and results of the digital transformation and other members of the team to perform the same analysis.	Recommendation
10. Prepare the next steps for digital pathology implementation after the workflow is well established.	Recommendation

## Data Availability

All the data used are present within the manuscript.

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
