# Peer review of "Best Practice Recommendations for the Implementation of a Digital Pathology Workflow in the Anatomic Pathology Laboratory by the European Society of Digital and Integrative Pathology (ESDIP)"

_diagnostics, 2021, doi:10.3390/diagnostics11112167_

Round 1
Reviewer 1 Report
The paper is interesting and can be considered for publication
However Authors should discuss some topics that are not included in the text
In particular:
- Describe the role of digital applications in dentistry, and the important advantages that determined such techniques. Please cite PubMed ID30769768;
- Digital applications can be particularly useful in parotid diagnostics PubMed ID19821124
- It's important to point out how digital techniques are useful in implant dentistry and orthodontics, and the same workflow can be applied to anatomic samples. Please cite PubMed ID25955953; PubMed ID26486206; DOI10.1177/1721727X1201000208
Author Response
Dear Editor, here are the replies to Reviewer 1.
Thanks for your comments and inputs. We are really glad you appreciated our work, we are sure that it will improve after the suggestions of the reviewers
(Comment) However Authors should discuss some topics that are not included in the text
In particular:
- Describe the role of digital applications in dentistry, and the important advantages that determined such techniques. Please cite PubMed ID30769768;
- Digital applications can be particularly useful in parotid diagnostics PubMed ID19821124
- It's important to point out how digital techniques are useful in implant dentistry and orthodontics, and the same workflow can be applied to anatomic samples. Please cite PubMed ID25955953; PubMed ID26486206; DOI10.1177/1721727X1201000208
Thanks for your precious suggestions. We are glad to stress that the digital revolution that is progressively changing pathology also affected or is affecting other disciplines, and thus we are glad to collect your inputs. For this reason, we selected the most appropriate reference among the ones kindly provided by you, and added the following sentence to make a parallelism in the heterogeneity of the digital devices available in both disciplines:
“6. Scanner for slide digitization:
[...As it happened in other medical specialties which have been dramatically changed by the introduction of a wide variety of digital devices for the routine daily work [30], the possibile scanner solutions for DP are numerous. Thus,..]”
Best regards,
Reviewer 2 Report
Major comments: The proposed guideline summarizes in detail current recommendations regarding digital pathology workflow. It contains very important and useful information not only for pathology departments how to proceed with all the steps of digital sample analysis. However, it is missing the part about digital data storage, its accessibility, and safety. I would suggest to include the paragraph describing how digital data should be stored, backed-up, and accessed in the way assuring its maximum safety.
Minor comments:
- Please increase the size of the font in the figure 1. Is is very hard to read.
- The title of the Table 4 should be "Scanning checkpoints".
Author Response
Dear Editor, here are our replies to Reviewer #2
Major comments: The proposed guideline summarizes in detail current recommendations regarding digital pathology workflow. It contains very important and useful information not only for pathology departments how to proceed with all the steps of digital sample analysis. However, it is missing the part about digital data storage, its accessibility, and safety. I would suggest to include the paragraph describing how digital data should be stored, backed-up, and accessed in the way assuring its maximum safety.
Thanks for your appreciation of our work. You perfectly stroke the point, and one of the most debated topics in the literature effectively regards these three main interrelated aspects, represented by the data storage (how and for how long to retain digital slides), storage solutions and data accession, all preserving the higher safety for both the specialists and the patients. We partly and briefly discussed this point in the section 8 (Open topics) of the original document, since no absolute and definitive indications still exist on these debated aspects. Anyway, we tried to further expand this section adding some more considerations on the details appropriately requested, hoping you and the other reviewer could appreciate the improvement of the paper with these further additions.
Minor comments:
- Please increase the size of the font in the figure 1. Is is very hard to read.
ADDRESSED
- The title of the Table 4 should be "Scanning checkpoints".
ADDRESSED